# Impacts of Adiposity on Exercise Performance in Horses

**DOI:** 10.3390/ani13040666

**Published:** 2023-02-14

**Authors:** Shannon Pratt-Phillips, Ahmad Munjizun

**Affiliations:** Department of Animal Science, North Carolina State University, Raleigh, NC 27695, USA

**Keywords:** equine, obesity, exercise, performance

## Abstract

**Simple Summary:**

Increased incidence of obesity in our equine population has clear negative impacts on equine health, such as increasing the risk of equine metabolic syndrome and laminitis. Excessive adipose tissue likely also has negative impacts on exercise performance, due to a combined inflammatory response and the effects of excessive weight carriage on work effort and limb health. This review explores research conducted in these areas.

**Abstract:**

There is ample research describing the increased risk of health concerns associated with equine obesity, including insulin dysregulation and laminitis. For athletes, the negative effect of weight carriage is well documented in racing thoroughbreds (i.e., handicapping with weight) and rider weight has been shown to impact the workload of ridden horses and to some degree their gait and movement. In many groups of competitive and athletic horses and ponies, obesity is still relatively common. Therefore, these animals not only are at risk of metabolic disease, but also must perform at a higher workload due to the weight of their adipose tissue. Excess body weight has been documented to affect gait quality, cause heat stress and is expected to hasten the incidence of arthritis development. Meanwhile, many equine event judges appear to favor the look of adiposity in competitive animals. This potentially rewards horses and ponies that are at higher risk of disease and reinforces the owner’s decisions to keep their animals fat. This is a welfare concern for these animals and is of grave concern for the equine industry.

## 1. Introduction 

The potential negative effects of extra weight carriage on a horse’s performance are not novel; handicapping of racehorses with additional weight was established in the 1800s. The Weight for Age scale was developed by Admiral Henry John Rous who suggested that older horses should be slowed down with weights when competing against younger horses [1]. Since those times, scientific research has indeed proven that carrying additional weight increases the workload of horses, as evidenced by slower race times and increased heart rates [2,3]. At the extreme, there are reports of welfare concerns regarding work efforts and strain on horses carrying excess weight (as riders or other loads) [4]. To this end, it would seem that additional weight carried by an overweight or obese horse as excess body fat would also contribute to towards a higher workload and potentially have a negative impact on athletic performance. The excess adipose tissue may also be detrimental to overall health and exercise performance directly through the production of inflammatory proteins and their effects on the body [5]. A large number of companion and performance horses and ponies are considered overweight or obese [6] and are, therefore, at risk of disease [7] and likely also have impaired performance. 

## 2. Adiposity in Horses

Adiposity refers to the extent of adipose tissue on the body. Obesity is characterized as a condition associated with having excessive body fat and/or being overweight. Defining obesity, therefore, not only requires the quantification of fat accumulation but also requires an establishment as to when body fat stores are considered “excessive”, which is having negative health consequences of having increased risk of disease [7,8,9]. There are two general methods of evaluating adiposity in horses: objective assessment and subjective assessment. 

Objective assessment includes carcass evaluation, which is the gold standard in adiposity assessment because it directly measures the amount of the adipose tissue through dissection [10,11], though clearly not useful in live animals. Other objective assessments include ultrasonography of subcutaneous fat [12,13] and measurement of total body water by the use of deuterium oxide [14] or bioelectric impedance [15]. Morphometric measures may also be useful as an objective measure of body size [16]. Subjective scores of body condition scoring (BCS) and cresty neck scores (CNS) have been developed to estimate fat coverage in horses [16,17]. 

Ultrasonography has been used extensively in horses to quantify body fat. Westervelt [12] described the use of ultrasonic measurements of rump-fat thickness to predict total body fat in ponies and horses. Authors have reported high correlation between tail-fat thickness with body condition score in Thoroughbreds, Quarter Horses and Arabians [18], while the fat tissue depth over the rump was best correlated with adiposity in Portuguese horses [19]. Ultrasonography has also been used to predict adiposity in donkeys [20]. Importantly, ultrasonography at 75% of the neck length (base of the neck) was correlated with parameters of metabolic disease, such as elevated plasma insulin and leptin concentrations [8].

The methods that quantify total body water to estimate adiposity are based on the principal that water is distributed in all parts of the body, except fat. By measuring either the distribution of doubly labeled water following a precision dose, or the movement of an electrical current across the body, the total body water and fat free mass can be determined. By difference, the fat mass can be calculated. Deuterium oxide has been validated as a measure of body fat in horses [14]. Similarly, bioelectric impedance has been assessed for use in horses, though it appears to overestimate fat mass [15]. 

Another novel system has been developed to objectively quantify body volume using 3-dimensional scanning methodology. This system could be used to monitor changes in volume due to either fat gain or loss, or muscle development or wasting. Torso volume was correlated with tailhead fat in Quarter horses [21]. This technology shows promise as it appears to be show high accuracy, requires no contact and is available at low cost [22].

Morphometric measures, such as girth circumference, abdominal circumference, neck crest height and neck circumference may also be used to assess adiposity [16]. In particular, girth: height ratio may indicate if animals are overweight or obese, by accounting for horse size. Such measures have also been used to describe adiposity in several types of horses and ponies and have been used to monitor weight change over time [8,23,24,25].

Subjective measures are also used to describe the subcutaneous fat coverage in an animal. A subjective body condition scoring scale of 1–9 has been widely used to describe the amount of fat coverage (or flesh) over a horse’s body [17]. Regions of the body examined include the tailhead, crease down the back, crest of the neck, withers, ribs and behind the shoulder. An emaciated horse would be assigned a score of 1/9, while a grossly fat animal would be assigned as score of 9/9, with a horse having a score of 5/9 being considered “ideal”. Kohnke modified the scoring system by scoring each region of the body and using the average score [26]. As Henneke applied this BCS system on Quarter horse broodmares, this system and the Kohnke modification have now been applied to a wider range of breeds (and classes) of horses such as Arabian racehorses [27], Thoroughbred geldings and mares [28,29], Icelandic and Warmblood horses [24,30], German Warmbloods [31] and ponies [10]. Body condition scoring has also been shown to be a repeatable measure of estimating adiposity [32]. Recognizing that some animals carry excessive fat in different regions of their body, Carter et al. also described the 0–5 Cresty Neck Score (CNS [16]). This system evaluates the fat deposition along the crest of the neck, where a score of 0/5 represents a horse with no palpable crest or fat accumulation above the nuchal ligament, a score of 3/5 represents a horse with an enlarged and thickened crest, and a mounded appearance, and a score of 5/5 describes a crest that is so large that it droops to one side. Both of these scales (BCS and CNS) have been used extensively in research and yet they are subjective scores and may not accurately quantify fat accumulation, particularly at higher levels of fat accumulation. 

Horses accumulate both subcutaneous fat and visceral fat by means of hypertrophy and hyperplasia of adipocytes. Extensive work by Dugdale and colleagues [10,14] characterized fat accumulation in carcasses of ponies and reported that both internal and external (subcutaneous) fat was equally distributed in the carcasses of thin through fat ponies. While the animals used in their study only ranged in BCS from 1.5–7/9 on the Henneke scale, it was evident that when body condition score increased above a score of about 6, there was an exponential increase in total adipose tissue. This documented that perhaps BCS was not a sensitive index of overall body fat particularly in fatter animals and favored the use of other morphometric measures such as retroperitoneal fat depth, which can be assessed by ultrasound, or even girth to height ratios. Martin-Rosset reported that the weight of fat tissues in horses was exponentially related to body condition score [33]. Similarly, Fowler et al. [28] suggested that BCS is a score for the overall adiposity of an equid rather than a score for a particular area of the body. This was confirmed, where despite not having significant correlations with fat thickness in each body area observed, BCS had a significant positive correlation with total body fat measured with D_2_O dilution [28]. Therefore, while not an objective measure of adiposity, the use of the body condition scoring system still has merit. 

EQUIFAT is a newer system to assess regional adipose tissue depots that was developed by Morrison and colleagues [34]. The assessment involves both subjective and objective gradings. The subjective grading is the assessment of fat depots from several regions of the body including the rump, epicardial, omental, and mesenteric areas, against specific descriptors (1 = no visibility of fat; 5 = excessive adiposity). Meanwhile, the objective grading is the assessment of the thickness scores of adipose tissue depots in nuchal crest and ventro-abdominal retroperitoneum, in which the depth range is assigned categorical scores ranging from 1 to 5. This method was developed to perform studies on the association between adiposity and the risk of some clinical conditions in live animals, during the surgery or in post-mortem conditions. 

Regardless of some potential limitations, and the availability of newer technologies, the Henneke body condition scoring of horses has been in place for numerous years and when coupled with cresty neck scoring, horses and ponies can be characterized as thin through obese. Using the Henneke scale, those horses scoring above a 6/9 are typically considered overweight and those scoring over a 7/9 being considered obese [6]. While body condition scores may not perfectly reflect the total white fat within animals, researchers have documented higher body condition scores with increasing risk of disease [7,35,36,37]. Similarly, research has shown that adjustments in dietary calorie intake can result in increases or decreases in body weight and body condition, likely as a result of adipose tissue gains or losses, respectively [25,29,38,39,40]. Based on several of these such studies, it has been estimated that one body condition score on an average 500 kg body weight horse, represents approximately 20–25 kg of fat [29,41,42,43]. 

## 3. Incidence of Adiposity 

Many studies have shown that upwards of 50% of the equine population is overweight, with 15–30% of those being classified as obese [44,45,46,47,48,49,50]. The number of overweight horses has been reported to reach 22–50% in the USA [44,45,46], and between 31.2–72% in the United Kingdom [48,50,51]. This high prevalence in obesity is likely attributed to lower work levels of many horses and increased access to high quality feeds, as well as a lack of understanding of how to assess adiposity [6].

Several studies have reported that owner perception of the level of adiposity has shifted, such that owners often do not recognize overweight or even obese animals, perhaps in part due to a lack of education [6,24,50,51,52]. In many of these studies, owners tend to underestimate their horses condition compared to a professional [23,24,52,53]. Another concern is the belief that show animals are expected to carry additional fat coverage in an effort to be more competitive in judged classes (such as conformation/halter classes, hunter and dressage disciplines), in particular compared to competitions that are not judged (show-jumping, eventing, polo, racing, etc.). Horse owners report that animals intended for the show ring were “more appropriate when overweight” [53]. Those authors surveyed horse owners about their ability to identify adiposity in a group of photos and found that those owners associated with showing horses were more likely to consider a healthy horse (BCS 5/9) as underweight. Furtado has conducted numerous studies regarding the relationship between obesity, horses and humans [54,55,56]. One study reports that owners describe “healthy horses” as having good coat condition and carrying sufficient fat. Owners also commented that fatter horses are “in show condition” and are “looking well” [54]. Incidence and concern of obesity in the show ring has also been reported in public media [57,58,59].

It has been shown that indeed, pony hunters do have higher body condition scores than polo ponies or show jumpers [60]. Similarly, it was reported that show and dressage horses were more likely to be overweight, and the incidence of obesity at one national show in the UK was 21% [48]. Another study documented that more than 80% of ponies competing at an international hunter competition were overweight, with 20% of them being considered obese [61]. Unfortunately, this study also documented that fatter animals were judged more favorably than leaner animals. To this end, a survey of hunter horse judges confirmed this notion by finding that that judges are more lenient towards overweight or obese horses, compared to those that might be slightly underweight when judged for conformation [62]. Seeing as fatter animals carrying more body condition are not healthy, such rewards for obesity will propagate the problem. 

Obesity may be more likely and understandable in leisure horses that may be overfed calories due to their lower level of activity and, therefore, lower caloric requirements. However as described above, overweight and obese horses have been reported in several athletic equine events, particularly those that are judged [48,60]. Studies report that judges may favor adiposity in sport horses [61,62], potentially influencing some competition horse owners and trainers to overfeed their animals on purpose [52,54]. The negative health consequences of obesity are well recognized, and yet many horse owners still keep their horses overconditioned in effort to give them an edge in the show ring. It is likely, however, that adiposity will ultimately not just shorten a horse’s lifespan, but may also negatively affect their athletic career [63]. 

## 4. Health Risks of Adiposity 

Adiposity in horses is linked to several equine clinical conditions including equine metabolic syndrome (EMS) [6,64,65], insulin dysregulation [66,67,68,69,70], adipose tissue dysfunction [68,71] and low-grade inflammation [5,72,73]. Obesity is also associated with susceptibility of developing laminitis [74,75,76] and reduced survival rate in laminitis cases [76,77,78]. Equine obesity was also found to cause accumulation of adipose tissue around key internal organs such as the kidneys and the heart, which may interfere with their functions [79,80]. Obesity also appears to affect reproduction in horses, in particular it appears to cause mares to continue cycling during the winter months, instead of going into anestrus [81], likely as a result of altered metabolic and other endocrine signals [82]. It should be noted that not all overweight animals are at risk of disease, though many are, particularly those with genetic tendencies or are coupled with higher dietary intakes of starch and sugar [6]. Many extensive reviews regarding the health consequences of equine metabolic syndrome and/or obesity are available [7,83]; a brief summary is presented herein. 

Many of these metabolic conditions stem from the understanding that adipose tissue functions to not only store energy reserves for the body, but also acts as an endocrine organ to secrete adipokines (compounds that regulate glucose and fat use and oxidation and metabolic processes) and cytokines (inflammatory regulators). There is mounting evidence that excess adipose tissue produces compounds that contribute towards conditions such as impaired insulin signaling and glucose dysregulation and the production of increased reactive oxygen species. To this end, basal insulin concentrations (a potential sign of insulin dysregulation) have been correlated to body condition score [46,84,85] in horses. Similarly, adipocytokine production is also related to body condition score and adiposity [73,85]. 

Equine Metabolic Syndrome, a disorder associated with insulin dysregulation, obesity and dyslipidemia is more prevalent in overweight horses, often culminating with the devastating hoof condition, laminitis. Laminitis is the inflammation of laminae, a delicate vascular system encapsulating the coffin bone within the hoof wall. The pathophysiology of laminitis is multifactorial, though its connection with insulin dysregulation stems from both a direct causation (wherein elevated insulin concentrations caused laminitis; [86]) and an association between increased basal insulin concentrations with increased lameness (pain) due to laminitis [87]. It is possible that obesity and insulin dysregulation, a heightened inflammatory state due to increased cytokine and adipokines, and perhaps along with other disorders such as Cushing’s syndrome, set up horses and ponies to be at greater risk of a laminitic episode, with a trigger factor (such as lush pasture or excessive carbohydrate ingestion resulting in a gluco-insulinemic response) eliciting the acute pro-inflammatory event [88,89]. Laminitis may also be caused by uneven weight bearing or work on hard surfaces (so-called road founder), suggesting that physical trauma and weight load to the laminae may also be damaging [90]. It is possible that added weight bearing in the limbs of overweight animals puts them at greater risk of developing laminitis [88,91], independent of any metabolic disorders and other risk factors. 

It stands to reason that perhaps not only the laminae may be strained by added weight carriage, but perhaps other structures within the lower limb, such as the navicular bone and bursa area or the suspensory ligament, as well as multiple joints are also strained. In our canine counterparts, it is well established that overweight animals are more likely to develop arthritis in their limbs, in part due to excessive weight forces on limbs over time [92,93]. In fact, in studies in which dogs are purposely underfed compared to ad libitum feeding, there is a delay in the onset and severity of arthritis [94,95]. The increased risk of arthritis with adiposity may be due to an influence of mechanical load on the joints, but may also be related to the proinflammatory compounds produced by the adipose tissue itself. In horses, Jaqueth found that arthritis was the most prevalent BCS-related disorder in overconditioned horses (31.8% [*n* = 512]) [96]. Of course, compromises to hoof, limb and joint health are increasingly detrimental to the career longevity of our athletic horses [63]. Therefore, feeding to keep horses at a leaner body condition may lengthen their athletic careers and prolong their lives. 

## 5. Increased Workload of Added Weight and/or Adipose Tissue

One consequence of adiposity on active horses is through the added workload due to increased weight carriage. As indicated, one body condition score represents approximately 20–25 kg of fat and, therefore, horses carrying “excess” body fat above an ideal body type (i.e., BCS of 5.0) could be carrying upwards of 50–100 kg of added weight (with BCS of 7–9/9). Therefore, studies have examined the effects of weight changes (as fat) on exercise performance, as well as the effects of carrying additional weight (as rider or other weights). These riding studies are relevant with respect to weight carriage, as Sloet van Oldruitenborgh-Oosterbaan reported no differences in workload between horses carrying a dead weight and an equal weighted rider [97]. 

In racehorses, there is a clear relationship between weight carriage and exercise performance. Thornton and Persson first demonstrated that an added 10% of body weight increased oxygen consumption by 15% in horses running on a treadmill [3]. Within racehorses, there also appears to be an impact of body fat weight on racing performance. Standardbred horses with a lower percentage of body fat reached a higher VO2_max_ (maximum oxygen consumption) [98] and reached a higher velocity for a given heart rate and plasma lactate concentration [2]. Similarly, work in elite Standardbred horses found that higher body fat content was associated with longer race times, that is, they were slower [99]. In Thoroughbred racehorses, race time increased significantly with bodyweight increases of more than 10 kg [100]. 

The impact of body fat on endurance racing may be more conflicting. Based on the aforementioned flat racing data presented above, carrying more weight may slow down an animal. However, for endurance rides, it is also important that the horse have sufficient fuel reserves to finish the race. Lawrence and others reported that the top seven finishers of a 241 km ride tended to have lower body condition scores and less rump fat than less competitive horses [101]. Meanwhile, Garlinghouse and Burrill reported that thinner horses (BCS < 3) were not able to successfully finish a 160 km endurance ride and that horses with higher body condition scores (between 5 and 5.5) had the highest completion rates [27]. 

Several studies have examined the impact of carried weight on the horse, generally due to the effects of the rider’s weight (Table 1). The effects of weight or load carriage brings into concern regarding welfare of not only sport horses but also working equids such as asses (for review see [102]. Criteria used to evaluate load capacity for working equids relates to gait symmetry and the ability to regain normal heart rate after work. In addition, markers of oxidative stress, lactate or cortisol concentrations may also suggest excessive workload that might be associated with poor welfare. The impact of weight carriage also poses challenges to riding schools if rider weight should be limited or accommodated based on horse size or type. Several studies have documented that overloading a horse (or other equid) may alter physiological parameters including heart rate, lactate and gait rhythm, as well as horse behavior. While the rider’s skill likely plays an important role in workload or welfare, the impact of weight carriage due to a rider is still relevant as it relates to weight carriage due to excess body fat. It should also be noted that these studies reflect an addition of weight at a point in time. In an obese horse, the added weight would contribute to lifelong changes in workload.


One study in Dutch Warmblood horses (BW 550–740 kg) tested the effects of additional loads in the form of a rider or being loaded with lead (both were 90 kg; approximately 15% of body weight) on the workload of the horses. Peak heart rates during the exercise, recovery heart rates after exercise and plasma lactate concentrations exercise were significantly higher in horses with both types of loads than in unloaded horses [97]. There was no difference in the parameters between the two types of loads. Stefánsdóttir et al. [103] compared the effects of different ratios between the weight loads and the horses on the horse workload. The weight loads consisted of a rider and additional loads such as saddle pads and extra heavy stirrups to make up 20%, 25%, 30%, and 35% body weight. They found linear increases in heart and respiration rates and exponential increases in lactate concentrations with the increase in the weight load ratio during a five-phase incremental exercise test. Work from our laboratory documented higher heart rates and increases in body temperature when horses were carrying an additional 15% of their body weight [104]. This study added weights to the horse’s backs via saddle, weight bags and feed bags attached to the horses, to remove any potential effects of the rider’s ability. Powell, et al. [105] compared horses carrying additional weight of four different percentages of bodyweight (15, 20, 25, and 30% of BW). These additional weights were in the form of a rider with different weights of tack. This research involved three different riders. Heart rate and respiration rate were higher in the horses carrying 25% and 30% BW, and lactate concentrations post exercise were higher in horses carrying 30% BW than the other groups. Dyson et al. [4] tested the effect of different rider weights (light, moderate, heavy, and very heavy) on equine welfare. Each rider performed a standardized dressage test on each of the five horses for 30 min. It was concluded that large riders could cause lameness and behaviors associated with musculoskeletal pain. However, Christensen et al. [106], tested the effects of additional 15% and 25% BW on horse behavior, physiological parameters, and gait symmetry and found that there were no short-term changes in the horse’s heart rate, cortisol or behavior with low intensity work. Wilk [107] also reported changes in body surface temperature with heavier loads. It is possible that both rider ability and rider body weight influence a horse’s workload and the effects on the horse may also be influenced by the type of work expected. animals-13-00666-t001_Table 1Table 1Effect of added weight on exercise related variables.ReferenceDetails% BW Added General Findings[106]Horses and ponies, dressage test15% RWNo change15% RWL25% RWL[4]Mixed breed horses, Dressage test 10–11.7% RWWeights > 15.3% BW induced temporary lameness and pain behavior 12.8–15% RW15.3–17.9% RW23.6–27.5% RW[104]Stock type horses, Light exercise in exerciser 0
15% L^ HR ^ RT[105]Light horses, Submaximal exercise test15% RW
20% RWL 
25% RWL^ HR, ^ RR30% RWL^ HR, ^ RR, ^ lactate [103]Icelandics, incremental exercise tests20% RWL

Linear ^ HR ^ RT 25% RWLExponential ^ lactate30% RWL 
35% RWL
[97]Dutch warmbloods, Treadmill exercise up to canter (7 m/s)0 ^ HR ^ lactate vs. no load~14% RW~14% L[107]Warmblood geldings, 33 min of walking and trotting10%
20%^ superficial body temperature^ = increased HR = heart rate, RR = respiratory rate, RT = rectal temperature. RW = rider weight; L = load (non rider weight), RWL = rider plus load weight.

While there are somewhat conflicting results regarding the impacts of individual riders, it can be concluded that weight carriage affects workload. As indicated, each additional body condition score may represent 20–25 kg of body weight (~5% BW of 500 kg horse) and, based on reports of many sport horses [60], some animals may be carrying more than 50 kg of added weight. Another study increased body weight through an increase in body fat content by feeding a high calorie diet. These horses had higher post-exercise heart rates and body temperature following incremental exercise on a treadmill [108]. Therefore, the impacts of added weight combined with the negative effects of adiposity tissue may significantly impact equine performance. 

## 6. Altered Movement and Kinematics Due to Added Weight and/or Adipose Tissue

In addition to increasing workload, added weight carriage may also impact a horse’s movement. The aforementioned study where horses were fed to gain fat weight also reported altered movement in these horses in addition to the effects on heart rate and temperature [108]. A study on riding school horses in Sweden found that there was a tendency that more asymmetric movements in the hindlimbs corresponded with higher BCS [109]. Moreover, asymmetry in both front limbs and hind limbs was higher in horses with higher BCS. 

Clayton reported that carrying an additional 18 kg of weight in a jumping exercise altered horse’s landing behavior and overall jumping kinematics [110]. This study used video recordings of horses carrying a rider (~61 kg) with or without an additional 18 kg of weighted cloth to determine the effects of weight on limb extension and stance durations upon landing from a 1.1 m high jump. The authors reported that the extra weight caused the landing forelimb to land closer to the fence and increased the extension of the carpal and fetlock joints of that limb [110]. Using force places, the ground reaction force (GRF) of a horse trotting in hand (not carrying weight) vs. carrying a rider were higher with the rider [111]. Jumping 0.8 m results in 3–5 times higher GRF than the canter, often with the trailing hindlimb carrying the highest loads [112]. 

Therefore, horses carrying heavier loads as added weight would likely have increased GRF on their limbs during flatwork and jumping. Walton reports that in canines, increasing body weight due to adiposity increases the mechanical force on joints and limbs, and contributes to the development of osteoarthritis [113]. 

The impact of additional weight on limbs can be approximated based on kinematic data from horses. Clayton and Hobbs [114] reported Peak Vertical Force (PVF) and Peak Breaking Force (PBF) in horses in units of Newtons/kg body weight. Ground reaction force describes the forces of the ground that would act against the hoof when it strikes the ground and can be measured using force plates or shoes, and includes vertical, longitudinal and transverse forces. Peak vertical force represents the maximum vertical forces when the hoof strikes the ground, and can be influenced by body weight, gait and speed, as well as surface footing and hoof/limb conformation. Breaking forces can describe the longitudinal forces to some degree, representing the energy directed to the leg from front to back. It can also be described as the force felt if an animal was moving and then suddenly stopped. Increases in PBF and PVF can increase the risk of injury [115]. For every body condition increase, there is an increase of approximately 25 kg, which would thus increase the PVF and PBF. Table 2 shows the estimated horse weights for an average 500 kg horse, with body condition scores of 4–9, with 5 being considered ideal at 500 kg. Table 2 also shows the increasing forces on the limb with increasing weight carriage. Based on this, a 550 kg horse that has a BCS of 7 (two BCS, ideal weight 500 kg) has an extra 722 N of force on their limb. It should be noted that horses that were naturally heavier due to breed and build would likely have heavier bone to support the extra force and weight. A smaller horse that is carrying extra load as fat may also have bone remodeling to support the body weight (as in humans [116]), but perhaps not sufficiently, and tendon support may be negatively impacted. Indeed, obesity is associated with tendinopathies in both humans [117] and dogs [118]. 

## 7. Other Consequences of Obesity 

Excessive adiposity and increased workload may also have other performance-limiting consequences. With muscular contraction, up to 60% of the energy metabolized is lost as heat (as opposed to being used for mechanical work) [119]. This heat is generally lost through evaporation, radiation, and convection. Evaporative cooling is the primary means by which horses dissipate heat, which requires blood flow to distribute heat to the skin surface, the production of sweat and its evaporation [120]. The insulating properties of fat, along with reduced surface area per kg of body weight, may contribute to heat stress in overweight animals. Indeed, it has been shown that horses with a BCS of 7.5 had a more difficult time dissipating heat in a hot/humid environment [121]. Another aforementioned study investigating the effects of rider weight on horses demonstrated higher superficial body temperatures in horses carrying heavier riders [107]. Heat stress is a major cause of retirement in many types of competition, including endurance racing, but may also compromise horses in other disciplines, particularly in hot and humid weather.

In humans, cardiovascular disease is a common comorbidity with obesity [122]. Similar associations between obesity and cardiovascular disease have been reported in canines and felines [123]. Siwinska reported architectural changes in the heart muscle of obese horses, as well as an increase in the pericardial and cardiac fat [80]. While cardiovascular disease is considered rare in horses [124], it is possible that more cases may be identified due to the incidence of obese horses. Additional research in this area is warranted. 

## 8. Conclusions

The data presented herein document a high incidence of obesity, particularly in some “show” disciplines. The increased weight carried (as fat) by such horses and ponies may be further increased when ridden, Such studies on the impact of rider weight carriage only investigated short term effects during the ridden workout. It has also been documented that adiposity is associated with several health concerns. It is likely that long-term adiposity and excessive weight carriage has additive effects over a horse’s life. Further research is required to document these consequences.

Despite the insurmountable recognition of major health consequences of obesity in our horses, several large research studies have shown that obesity is prevalent among our equine populations. Excessive weight carriage—as body fat and/or rider weight—appears to increase the workload of horses and may pose excessive strain on a horse’s limbs during exercise. Excessive weight carriage as fat further increases the likelihood of additional negative health impacts such as metabolic disease, laminitis, inflammation, cardiovascular changes and/or arthritis. Therefore, excessive adiposity may contribute to potentially shortened careers and/or lives of these equine partners. The current trends in many equine disciplines that appear to reward excess condition (fat) and the increased overall incidence of obesity in our equine populations is of grave concern. 

## Figures and Tables

**Table 2 animals-13-00666-t002:** Estimated forces on limbs with horses of varying body condition scores and thus body weights.

Horse Weight	BCS	Walk		Trot		Canter/Jump *	
	-	Peak Vertical Force (N)	Peaking Breaking Force (N)	Peak Vertical Force (N)	Peaking Breaking Force (N)	Peak Vertical Force (N)	Peaking Breaking Force (N)
	-	4.45 N/kg	−0.80 N/kg	10.52 N/kg	−1.05 N/kg	14.45 N/kg	−1.35 N/kg
475	4	2113.8	−380.0	4997.0	−498.8	6863.8	−641.3
500	5	2225.0	−400.0	5260.0	−525.0	7225.0	−675.0
525	6	2336.3	−420.0	5523.0	−551.3	7586.3	−708.8
550	7	2447.5	−440.0	5786.0	−577.5	7947.5	−742.5
575	8	2558.8	−460.0	6049.0	−603.8	8308.8	−776.3
600	9	2670.0	−480.0	6312.0	−630.0	8670.0	−810.0

Adapted from Clayton and Hobbs [114], * Trailing forelimb, Jump < 0.8 m.

## Data Availability

Not applicable.

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
