# Peer review of "Impacts of Adiposity on Exercise Performance in Horses"

_animals, 2023, doi:10.3390/ani13040666_

Round 1
Reviewer 1 Report
The topic is interesting because overweight/obesity is a current problem in horses around the world.
The review should be focused on the impact of adiposity in exercise performance in horses, not mainly on the consequences of having more weight carriage, as rider weight or additional load weight.
For a review article that is intended to summarize the available scientific work on the effect of adiposity (not raider or load weight) on performance of horses, the document should include more information about that. Otherwise, should be rearranged to present the physical/health consequences of different loads (rider or additional load) over the horse.
Also, conclusions should conclude (worth the redundancy) the main effect of obese horses (and adiposity) on performance.
Keywords are necessary.
Check reference citations, all should be the same.
Author Response
Thank you for your comments.
We realize that fat weight is very different from rider weight - but there are also far more studies on rider weight and their effects on workload, than there are simply on the effects of fat weight on workload. Also - a few studies have compared the effects of rider weight vs. a similar dead weight - and found no/few differences. We have tried to emphasize this point further.
We also tried to not focus solely on general health consequences of obesity - as there are already several excellent reviews on the topic. We tried to focus more so on the impacts with exercise - but have expanded this area.
Reviewer 2 Report
Overall, this is a well written review paper on the impacts of adiposity on exercise performance. However, there are many instances of entirely missing or very limited references. As this is a review article, it is essential that the authors ensure that information is appropriately referenced and that all relevant information is referenced.
L29: Please include additional information on these two methods.
L43: Please provide further details on CNS
L85: Consider adding information on 3D photonic scans to to this section (Valberg et al. 202, Matsuura et al. 2021).
Also relevant to this section are Silva et al. 2012, Martin-Gimenez et al. 2016, Webb & Weaver 1979, Carroll & Huntington 1988, Kienzle & Schramme 2004, Martin et al. 2008, Carter & Dugdale 2013, Silva et al. 2012, Superchi et al. 2014, Gentry et al. 2004, Quaresma et al. 2013, Westervelt et al. 1976, Motett et al. 2009, Gobesso et al. 2012, and others.
L87: Should be "the USA"
L91: References are associated with what?
L92: unclear
L95: Please expand this section on owner/trainer/judge perception of weight and its impact
L112: interfere WITH their functions
L150: In horses, Jaqueth et al. (2018) FOUND THAT arthritis...
L156: Add Fonseca et al. 2013
L158: ...due to INCREASED weight carriage
L169: define VO2max
L211: Explain how these kinematics were altered
L238: ...horses in OTHER DISCIPLINES.
L239: Many new points are introduced in this section. Consider moving this information up to the Incidence section. Then rewrite the conclusion to act as a summary of all sections
Author Response
Thank you very much for your careful review - and your suggested references. We have incorporated most of those where applicable.
We have also expanded on other sections where you suggested.
Again, thank you for these comments and for identifying the grammar errors and extra references!
Reviewer 3 Report
Thank you for contributing to creating an overview over a complex, ubiquitous and important topic.
Please add references for statements in line 9-19, 44-46, 100-101, 138-143, 224-225, 226-229
Also reference for 258-259 (conclusion) should be available further up in the text
Line 94-97: please refer to particular disciplines/level of competition that are related to favoritism of adiposity in athletic use of horses, as there is a huge difference in between, and some disciplines seems to be a protective factor against over-weight for sport horses (Is it for example only "Showing" you refer to?) -it is described in more details in the conclusion; please describe the facts/details before the conclusion
Please define the term "athletic horse", for example used in line 157: is it equal to riding horse in general or specifically a sport, race or competition horse?
Line 117: Is "obese adipose tissue" a recognised term? - or should it be excess adipose tissue in the case of obesity?
Line 181-204: Please indicate which of the weight studies includes changes of riders, as this is known to potentially cause bias in the results. Also, please indicate how the horses in the studies have been weighed/scored as part of the assessment (horse/rider weight ratio). Are the included horses of ideal weight?
Please include all of the mentioned references in table 1 for comparison (also Christensen), perhaps also include whether there has been a change of rider in the study design, as this is important.
261-264: Can this be understood as excessive weight due to adipose tissue is worse than due to for example rider weight? Please add a reference then (or perhaps just rephrase the sentence)
Overall: is it unequivocally shown that increase in weight/weight carriage is always problematic for riding horses/horses? (at which level of use) Or is it only indicated by studies that it can/might be, but perhaps with some important challenges in the study designs made so far? Do we need further understanding of what adipose tissue and overweight does to horse health? What are facts and what are assumptions?
Author Response
Thank you for your careful review of our article.
Please add references for statements in line 9-19, 44-46, 100-101, 138-143, 224-225, 226-229
- These have been added where appropriate. Some ideas are the authors own knowledge.
Also reference for 258-259 (conclusion) should be available further up in the text
- this area has been changed signficantly
Line 94-97: please refer to particular disciplines/level of competition that are related to favoritism of adiposity in athletic use of horses, as there is a huge difference in between, and some disciplines seems to be a protective factor against over-weight for sport horses (Is it for example only "Showing" you refer to?) -it is described in more details in the conclusion; please describe the facts/details before the conclusion
- more detail has been added here
Please define the term "athletic horse", for example used in line 157: is it equal to riding horse in general or specifically a sport, race or competition horse?
- thank you - this has been replaced
Line 117: Is "obese adipose tissue" a recognised term? - or should it be excess adipose tissue in the case of obesity?
- excess adipose tissue! Thank you for the catch!
Line 181-204: Please indicate which of the weight studies includes changes of riders, as this is known to potentially cause bias in the results. Also, please indicate how the horses in the studies have been weighed/scored as part of the assessment (horse/rider weight ratio). Are the included horses of ideal weight?
- we have revise this section and the table. Many studies have multiple sizes of horses, and some do not include body condition score. We chose to reflect weight added as a % of the horse's body weight.
Please include all of the mentioned references in table 1 for comparison (also Christensen), perhaps also include whether there has been a change of rider in the study design, as this is important.
- thank you, we have added to the table to show if it was load or rider or both.
261-264: Can this be understood as excessive weight due to adipose tissue is worse than due to for example rider weight? Please add a reference then (or perhaps just rephrase the sentence)
- yes, thank you. We have rephrased this area
Overall: is it unequivocally shown that increase in weight/weight carriage is always problematic for riding horses/horses? (at which level of use) Or is it only indicated by studies that it can/might be, but perhaps with some important challenges in the study designs made so far? Do we need further understanding of what adipose tissue and overweight does to horse health? What are facts and what are assumptions?
- Thank you. This is an important concept that we have tried to include. The studies looking at rider weight are only a point in time, rather than a potential lifetime as in an overweight horse. More research is clearly needed to examine these long term effects.
Round 2
Reviewer 1 Report
The title should consider additional weight (not only adiposity) considering the review is about both additional weight and adiposity.
Keywords are needed
L196: some reproductive effects should also be mentioned.
L120: Fowler et al. (28)
L154: access or assess?
L166-167: should cite some of Furtado´s studies, otherwise that phrase is out of context.
L281: (for review see 106)
It would be interesting to add more of the effect of obesity and consequences with additional weight over horse welfare.
Author Response
Thank you for your continued support.
I put in keywords and then they are not in the manuscript! I will try again.
A statement regarding the effects of obesity on cycling in mares has been added.
What you have been sent doesn't appear to be what I submitted - with respect to the Furtado reference. I will need to track down why it was changed.
This is what is currently written - with Furtado as reference 56:
It has been shown that indeed, pony hunters do have higher body condition scores than polo ponies or show jumpers [62]. Similarly, it was reported that show and dressage horses were more likely to be overweight, and the incidence of obesity at one national show in the UK was 21% [63]. Another study documented that more than 80% of ponies competing at an international hunter competition were overweight, with 20% of them being considered obese [64]. Unfortunately, this study also documented that fatter animals were judged more favorably than leaner animals. To this end, a survey of hunter horse judges confirmed this notion by reporting that that judges are more lenient towards overweight or obese horses, compared to those that might be slightly underweight when judged for conformation [65]. Seeing as fatter animals carrying more condition are not healthy, such rewards for obesity will propagate the problem.
Obesity may be more likely and understandable in leisure horses that may be overfed calories due to their lower level of activity and therefore lower calorie requirements. However as described above, overweight and obese horses have been reported in several athletic equine events, particularly those that are judged [62, 63] Studies report that judges may favor adiposity in sport horses [64, 65], potentially influencing some competition horse owners and trainers to overfeed their animals on purpose [54, 56]. The negative health consequences of obesity are well recognized, and yet many horse owners still keep their horses overconditioned in effort to give them an edge in the show ring. It is likely however that adiposity will ultimately not just shorten a horse’s lifespan, but may also negatively affect their athletic career [66].
I have also added to the comments about welfare in working equids/horses.
Reviewer 2 Report
The manuscript has been significantly improved and only requires editorial review.
Author Response
Thank you very much!
Reviewer 3 Report
Thanks for the revised version
Author Response
Thank you!
Round 3
Reviewer 1 Report
The title should be modified to better reflect the content of the document...as a suggestion, could be something like Obesity and impact of adiposity/additional weight on performance in horses.
Keywords are not in the document, but maybe will be in the final version?